# Wood structure explained by complex spatial source-sink interactions

Andrew D. Friend [1] ✉, Annemarie H. Eckes-Shephard [2] & Quinten Tupker [3]

Wood is a remarkable material with great cultural, economic, and biogeo-chemical importance. However, our understanding of its formation is poor. Key properties that have not been explained include the anatomy of growth rings (with consistent transitions from low-density earlywood to high density latewood), strong temperature-dependence of latewood density (used for historical temperature reconstructions), the regulation of cell size, and overall growth-temperature relationships in conifer and ring-porous tree species. We have developed a theoretical framework based on observations on *Pinus sylvestris* L. in northern Sweden. The observed anatomical properties emerge from our framework as a consequence of interactions in time and space between the production of new cells, the dynamics of developmental zone widths, and the distribution of carbohydrates across the developing wood. Here we find that the diffusion of carbohydrates is critical to determining final ring anatomy, potentially overturning current understanding of how wood formation responds to environmental variability and transforming our interpretation of tree rings as proxies of past climates.

Lignified cellulose, "wood", is the largest store of carbon in terrestrial plants, and as such plays a major role in the global carbon cycle[1]. Understanding controls on its formation is a major research priority[2,3]. Unlike most metabolic pathways, the process of wood formation, xylogenesis, is a synthetic dead-end without the potential to recycle materials, and it is likely for this reason that it is highly regulated, avoiding wasteful allocation of photosynthate[4]. This regulation results in conserved anatomical patterns, with tree rings in conifers and ring-porous species exhibiting transitions from low to high density wood each year in extra-tropical regions. Nevertheless, these rings still carry the imprint of environmental conditions, a feature exploited in climate reconstructions, and as such underpin our ability to place contemporary climate change in an historical perspective[5,6]. However, despite the importance of this imprint for understanding past climates, and the relevance of wood formation for carbon sequestration and tree physiology, our understanding of how tree-ring anatomy arises, and how its variability at both intra- and inter-annual timescales is controlled, remains poor. This is exemplified by our inability to

explain the so-called "divergence problem", the reduction in temperature sensitivity of tree-ring proxies in recent decades, resulting in significant theoretical and practical consequences for the use of tree rings in palaeoclimate reconstructions and for the quantification of global warming trends[7].

A number of approaches to modelling wood formation have been published, and were recently reviewed[8]. These models have mostly targeted a subset of drivers and processes and usually aimed to predict a single wood property (e.g. density, ring width, or cell number). As such they have not to date provided a general mechanistic explanation of a complete set of fundamental wood properties and dynamics. For example, the first computer model of wood formation treated the passage of cells through cambial, enlarging, and secondary wall-thickening phases, but did so using a descriptive approach with no environmental factors (Howard and Wilson)[9]. These were introduced in the widely used Vaganov-Shashkin (VS) model[10], which is focussed on tree growth responses at climatic limits in order to interpret climate signals (especially temperature) for dendroclimatological

[1]Department of Geography, University of Cambridge, Downing Place, Cambridge CB2 3EN, UK. [2]Department of Physical Geography and Ecosystem Science, Lund University, Lund S-223 62, Sweden. [3]Department of Pure Mathematics and Mathematical Statistics, University of Cambridge, Wilberforce Road, Cambridge CB3 0WB, UK. ✉e-mail: adf10@cam.ac.uk

applications. It uses a limiting factor (water, temperature, or day-length/carbon) for cell production, but it is not clear if it is applicable outside of regions with strong climatic controls. Other models have focussed on intra-ring anatomy by resolving the most limiting environmental factor on each developmental stage (Deleuze and Houllier)[11], cellular dynamics and hormonal signalling, but with no environmental responses (Hartmann et al.)[12], intra-tree carbon dynamics using an empirical approach to growth (Schiestl-Aalto et al.)[13], and carbon-water interactions using very detailed hydrological dynamics (Hölttä et al.)[14]. Other models have investigated environmental effects on cell production rates (Cabon et al.)[15] and carbon effects on cellular development (Cartenì et al.)[16]. Finally, Drew et al.[17] developed a complex model to investigate wood properties in *Eucalyptus* based on the diffusion of an hormonal signal and incorporating environmental controls. Alongside TreeRing 3 (Fritts et al.)[18], their model is one of the few to have the potential to simulate a large variety of ring growth dynamics and key anatomical features, in response to both environmental and internal drivers, but the structures and assumptions of these two models have never been interrogated using model experiments. Overall these, and other models[8], simulate single phenomena and fit well local observations related to wood formation dynamics, ring width, or anatomy. However, they are surprisingly diverse in their underlying assumptions, including the influences of hormonal and environmental drivers. Therefore there remains considerable uncertainty regarding our understanding of fundamental wood formation processes and how to model them. Moreover, existing approaches have not been simultaneously tested across a range of key anatomical features. In addressing multiple questions related to different processes within a single approach, we present in this paper a general framework that is comprehensive, robust, biologically plausible, and skillful in reproducing observed phenomena.

Here we describe a theoretical framework and use it to investigate how wood formation processes result in final ring structures under temperature-limited conditions. We combine insights from (i) studies on wood formation at the intra-annual timescale[19–21], (ii) a theoretical study of cell-size and growth regulation in plant meristems[22], and (iii) high-resolution characterisation of biochemical and anatomical properties during wood development[23]. Our framework explicitly considers xylogenesis in time and space, and thereby provides a mechanistic basis for understanding and predicting how the rate at which carbon is sequestered into wood and wood structure vary with environmental conditions at daily to inter-annual timescales. We focus here on the effects of temperature as this is the dominant driver of inter- and intra-annual variability in wood formation in many temperate and boreal locations, with consequences for ring width, mass, and maximum density, and for which key scientific questions about these responses remain unanswered[21]. However, our framework is structured in such a way that other environmental drivers of wood formation, such as soil moisture and nutrient levels, can be incorporated through controls on the rate of cell enlargement, which in turn affects cell proliferation (see below).

*Pinus sylvestris* L. (Scots pine) is used as the model species here due to the good availability of key observations with which to develop and test our simulation model. As in most conifers, its growth rings typically consist of earlywood with large cells and thin secondary walls, which transitions to latewood with small cells and thick walls in the final third or so of the ring. The total annual carbon increment is the integration of this density distribution over the total ring width, and cell lumen volumes and wall thicknesses determine wood properties such as conductivity to water and mechanical strength. However, the mechanisms that create this pattern are not understood[21]. Recent observations of intra-annual cellular kinetics have quantified the relative contributions of different processes, such as cellular enlargement[19,21,24,25], but have not yet been integrated into a mechanistic explanation of how the final structures are created. A key question

is why cellular mass density is relatively unresponsive to temperature, except in the final part of the developing ring. A paradigm that has emerged from statistical analysis of intra-annual observations is that the lack of temperature sensitivity in earlywood is due to compensation mechanisms between rates and durations, which seem to break down in latewood[26], resulting in maximum latewood density being an important proxy for past climate[6].

The framework explored here combines, in a numerical model, observations, recent advances in understanding, and hypotheses for the fundamental mechanisms that underlie xylogenesis. The model, fully described in the Methods, computes the daily development of individual cells arranged along a specified number of independent radial files (Fig. 1). It takes as inputs latitude and daily air temperature. Its initial condition is the state of cells within the cambium (the "proliferation zone"), and its physical domain is from the cambial initial cell to the innermost living cell along each radial file. The model steps through a prescribed number of years and within each year computes daily changes in the state of each cell, which consists of radial position relative to the innermost edge of the phloem, radial length (axial and tangential lengths are assumed fixed), the concentration of a division-inhibiting molecule in the lumen, mass (increased by the laying down of primary and secondary wall material), lumen carbohydrate concentration, and whether it is dormant, active, or dead. Proliferating cells divide periclinally when they reach a critical size. A cell's radial position evolves in time as a function of the number of cells and their lengths in the radial file between the phloem and the cell. An individual cell's carbohydrate concentration is determined from its position relative to the phloem, the demand for wall growth across the developing radial file, and the (constant) carbohydrate concentration in the phloem. Superimposed across all radial files is a spatial

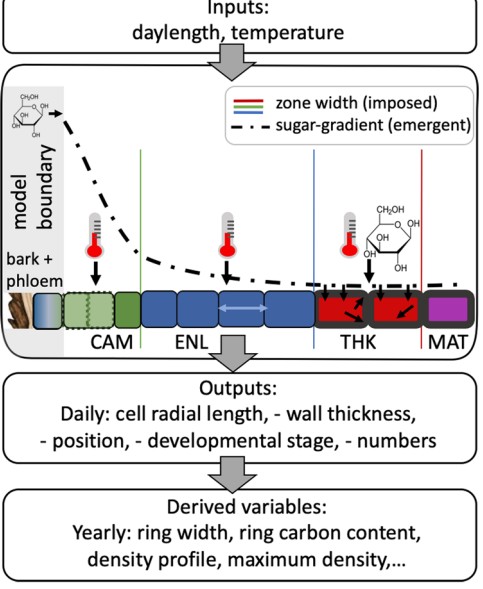

**Fig. 1 | Model scheme of inputs and outputs alongside one representative radial file (shown during the growing season when all cell types are present).** CAM cambium or proliferation zone, ENL enlargement-only zone, THK secondary wall-thickening zone, MAT mature zone with dead cells. The zones are delineated by zone widths (coloured vertical lines). Cells within each zone can only undergo certain processes (e.g. division in the proliferation zone, secondary wall thickening in the thickening zone). Temperature (thermometer icon) acts on all living cell types. A sugar-gradient (dot-dash line) emerges as a function of phloem sugar concentration and the requirements for primary wall growth of enlarging cells and secondary wall thickening of thickening cells. Derived variables are calculated from the model outputs.

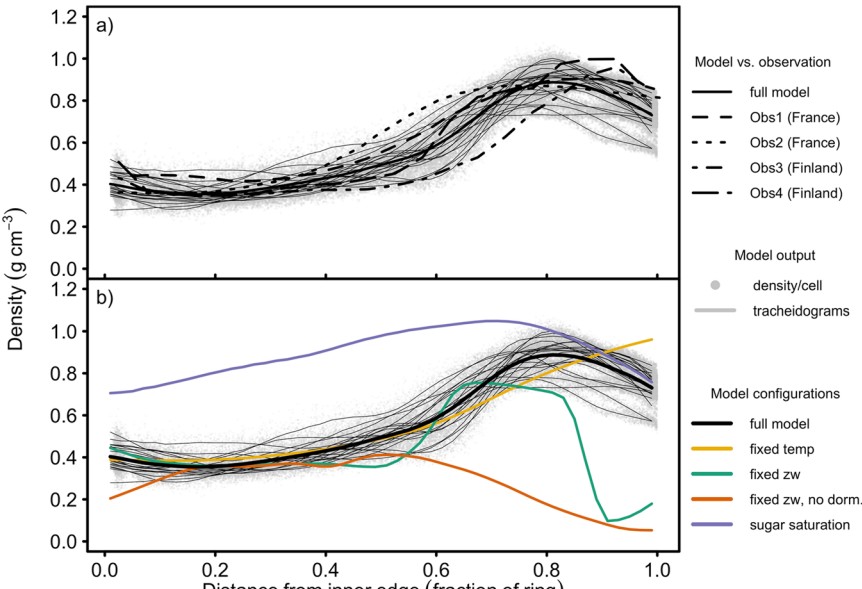

**Fig. 2 | Simulated cell-mass densities (i.e. cell mass/volume) across normalised ring width of *Pinus sylvestris* L. at 64.25°N, 19.75°E (northern Sweden), 1976–1995.** Simulations used a spin-up over 1951–1975, with proliferating cells carried over between years. Light grey solid lines are annual mean profiles of simulated cell-mass density standardised across 100 independent radial files per simulation. Wider solid lines (representing different model configurations) are the mean of all annual cell-mass density profiles (light grey lines in the case of the default simulation) across the entire simulation period. The mean profiles were calculated by binning cells in 0.02-fraction-wide bins and then computing the running mean over three adjacent bins. **a** Comparison with observations of the same species from: France, morphometric density (Obs1)[19]; France, X-ray (Obs2)[19]; southern Finland, X-ray (Obs3)[47]; and northern Finland, X-ray (Obs4)[47]. **b** Effects of changing model assumptions on density. Fixed temperature ("fixed temp"): 10 °C for cell enlargement and wall thickening; fixed zone widths ("fixed zw"): developmental zones always at their 20th June values, with or without autumn dormancy; sugar saturation assumes $\Delta M = \Delta M_{max}$ (Eq. (8)). Individual annual simulation outputs are shown in Supplementary Fig. 1.

configuration of three developmental zones which determine each cell's developmental phase. These proliferating, enlargement-only, and secondary wall thickening-only zones vary in width (defined as distance from the phloem) over days of the year, but do not vary between years or radial files. These zones determine the states that can change for cells within them (e.g. radial length can only change within the proliferating and enlargement-only zones). At the end of a simulation year cambial cells are carried over to the new year, with other cells added to the sapwood as a new annual growth ring.

The model incorporates the following assumptions within this overall framework: (i) when not dormant, a single bifacial initial (stem) cell in each radial file grows radially and undergoes periclinal divisions, producing phloem outwards and xylem inwards[27]; (ii) these divisions are stochastically assigned to xylem or phloem, with prescribed probabilities[28]; (iii) when not dormant, inward cells can grow radially if in the proliferation or enlargement-only zones, and can divide if in the proliferation zone, with division-size regulation intermediate between a constant size and constant increment[22] (this assumes that the vascular cambium and the apical meristem share regulatory mechanisms[29]); (iv) division is asymmetrical, with the relative growth rates of the daughters conditioned on their initial relative sizes, maintaining homeostasis[22]; (v) initial cells and their derivatives elongate in the radial direction at a rate dependent on temperature[30] (calibrated to[31]; other factors such as turgor and nutrients are not considered here; the baseline elongation rate is calibrated to[23]); (vi) once cells are no longer in the proliferation zone they continue to enlarge at the temperature-dependent rate while in an enlargement-only zone; (vii) once they are no longer in the enlargement-only zone, secondary wall thickening can occur until they are no longer in the wall thickening-only zone, at which point they are assumed to lose their protoplasm and become mature, functioning, xylem; (viii) the rates of primary (within the proliferating and enlargement-only zones) and secondary wall growth depend on temperature (calibrated to[31]), and

the local availability of carbohydrates (calibrated to[23]); (ix) carbohydrates diffuse from the phloem along each radial file, with a prescribed (constant) phloem concentration and constant resistance between cells, creating a new equilibrium profile each day as a consequence of the balance between carbohydrate supply and demand at the cellular level (calibrated to[23]); (x) the widths of the proliferation, enlargement-only, and wall thickening-only developmental zones vary as functions of daylength (calibrated to[23]); (xi) the proliferation and enlargement-only zones enter dormancy at a prescribed daylength[32], and exit dormancy when a chilling day-conditioned heat sum is reached[33] (calibrated to[23]).

## Results
### Overall density profile

The simulated distribution of cell-mass density across the growth ring at the end of the growing season compares favourably to observations of this profile in Scots pine from the literature (Fig. 2a), suggesting that the model's assumptions are plausible. No model parameter or underlying assumption was adjusted in order to match any observation of density distribution - the predicted pattern and the absolute values are an emergent property of the model. In both observations and simulations, density remains low and fairly constant at ca. 0.4 g cm⁻³ for the first 40–50% of the ring, and then increases to a mean peak of ca. 0.9 g cm⁻³ in the final 10–30% of the ring, before declining in the final 10–20%.

The mechanisms responsible for the observed intra-annual density pattern are not known, but could involve, singly or in combination, climate seasonality (e.g. through a direct effect of temperature and/or water availability on cellular development), changing developmental zone widths and/or activity (possibly related to daylength signalling), and/or carbon supply (possibly related to variability in photosynthesis and/or whole-plant source/sink dynamics). The potential mechanisms were examined using our model. Removing the effect of temperature

variability on both cell enlargement and wall thickening does not change the overall pattern of low-density earlywood transitioning to high-density latewood, but does smooth the rate at which density increases towards the latewood, and removes the density decline in the final 20% of the ring (Fig. 2b, "fixed temp"). Therefore, while temperature seasonality is not responsible for the overall density pattern in the model, some features, such as the existence of a peak in the latewood, are temperature-controlled. Furthermore, we found that the late-season peak also disappears when the effect of temperature on wall thickening alone is removed, demonstrating that in the full model this peak is caused by low late-season temperatures reducing the rate of wall thickening.

The effect of changing zone widths over the year was investigated by fixing all widths at their 20th June values (Fig. 2b, "fixed zw"). The resulting profile still has low-density earlywood and an increase in density in latewood, suggesting that zone width variability is not the primary cause of the early-latewood transition. However, in contrast to the full model, the transition is steeper and there is a rapid decline in density at ca. 85% of the ring. Analysis of cellular dynamics through the year (cf. Supplementary Fig. 2) shows that this late-season decline is due to cells stuck in the enlargement-only zone when dormancy occurs in autumn. These cells do not continue to enlarge under this scenario, nor does the enlargement-only zone width decline, and so they do not enter the thickening zone and therefore never undergo any secondary wall thickening. Removing autumn dormancy in addition to keeping the zone widths fixed produces very wide rings because many additional cells are produced and enlarge, but there is no longer any high-density latewood (Fig. 2b, "fixed zw, no dorm."). Therefore although zone width variability per se is not the cause of the earlywood-latewood transition, the cessation of enlargement due to late-season dormancy is required, increasing the duration of wall thickening relative to enlarging.

The influence of carbohydrates on the density profile was investigated by allowing wall thickening to continue at its maximum, temperature-limited, rate, with no constraint from the supply of carbohydrates (Fig. 2b, "sugar saturation"). While there is still an increase in density across most of the profile, and a peak in the latewood, earlywood has a much higher density than in the full model and the earlywood-latewood transition is smooth. The gradual increase in density across the earlywood with saturating carbohydrates is due to the seasonal narrowing of the enlargement-only zone, reducing cell size (note that density is constant across the first ca. 50% of the ring if zone widths do not change but still increases if temperature is fixed). The low mass density in earlywood cells in the full model is therefore due to low carbohydrate concentrations during the wall thickening phase. These low concentrations are in turn the result of the greater total resistance to carbohydrate diffusion from the phloem to the wall-thickening zone during the early growing season due to the greater number of intervening enlarging cells (cf. Supplementary Fig. 2). The densities across the latewood when carbohydrates are saturating and in the full model converge towards the end of the ring (Fig. 2b). The reason for this convergence is that in the full model late-season cells undergo wall thickening closer to the phloem than cells maturing earlier, resulting in lower carbohydrate limitation. This allows these cells to deposit more carbon in their walls, increasing the density of latewood relative to earlywood.

Taken together, these simulations demonstrate that the overall density profile in the tree ring, and hence its final carbon content, is mainly determined by the interaction of the spatial distribution of carbohydrates across the radial files, the contraction of the enlargement-only zone during the growing season, and the imposition of autumn dormancy on cell enlargement. The spatial distribution of carbohydrates between cells (shown in Supplementary Fig. 5) is an emergent property of the developing ring, determined by the concentration in the phloem and the activities of the proliferation, enlargement-only, and wall thickening zones. Wall thickening is in turn determined by the local concentration of carbohydrates, resulting in a complex spatial interaction between sources and sinks.

## Temperature and cell-mass density

Our model was used to investigate possible mechanisms for the observed high sensitivity of cell-mass density to temperature within latewood[26]. Adding a fixed temperature increment of 2 K to daily temperature (while keeping dormancy forcing unchanged) results in a wider radial file, with the increase due to more earlywood (Fig. 3), as is observed in diverse tree species with respect to the relationship between ring width and the fraction of earlywood[34,35]. Mean earlywood density is slightly reduced with increased temperature, whereas latewood density is increased strongly, consistent with observations[36]. Applying the temperature increment to wall thickening but not enlargement results in almost no change in earlywood density from the default simulation (Fig. 3), showing that the reduction in earlywood density with increased temperature is due to its effect on cell elongation, leading to cell size increasing proportionally more than mass. However, latewood density is increased to the same extent as when temperature is applied to both thickening and enlarging, showing that the latewood density temperature response is due to the effect on thickening alone. The control on earlywood density from cell enlargement and latewood density from cell wall thickening in our model is consistent with anatomical analysis in different species across a broad range of locations[36].

The simulated sensitivity of density to temperature across the radial profile is consistent with the paradigm of compensation mechanisms occurring in the earlywood which break down in the latewood[26]. The compensation mechanism has remained elusive, but has been postulated to be a consequence of tight negative coupling between rates and durations during xylogenesis[21]. However, our finding that the high sensitivity of density to temperature occurs throughout the ring under conditions of saturating carbohydrates (Fig. 3b) challenges this view. The low to negative effect of temperature on density in the earlywood of the full model is attributable not to compensation between rate and duration, but rather to the limited supply of carbohydrates, similarly to how carbohydrate supply controls absolute density (cf. Fig. 2b). In contrast, high concentrations of carbohydrates in the latewood, due to its proximity to the phloem during wall thickening (see Supplementary Fig. 2), mean that wall building is relatively more limited by the temperature kinetics of the biosynthetic machinery (Eq. (9)) than the substrate concentration (Eq. (8)). The cellular and zone-width dynamics responsible for the final ring anatomy are shown in Supplementary Fig. 2. Earlywood cells spend much less time in the wall thickening zone than latewood cells due to a high rate of proliferation and the large width of the enlargement-only zone, quickly pushing the phloem outwards as the radial file lengthens. As the enlargement-only zone narrows over the growing season, cells spend increasing amounts of time in the wall thickening zone and, crucially, have fewer cells between themselves and the phloem, thereby experiencing higher carbohydrate levels. Greater supply of carbohydrate increases the role temperature plays in the rate of wall deposition in latewood cells; the cell walls of the latewood cells therefore become imprinted with temperature signals, leading to their utility as a climate proxy.

## Temperature and cell size

Increased temperatures lead to greater rates of cell enlargement and hence proliferation in our model. However, temperature increase results in a slight reduction in cell size across much of the ring (Fig. 4), demonstrating the existence of strong compensation mechanisms. The relative roles of cell enlargement versus proliferation in

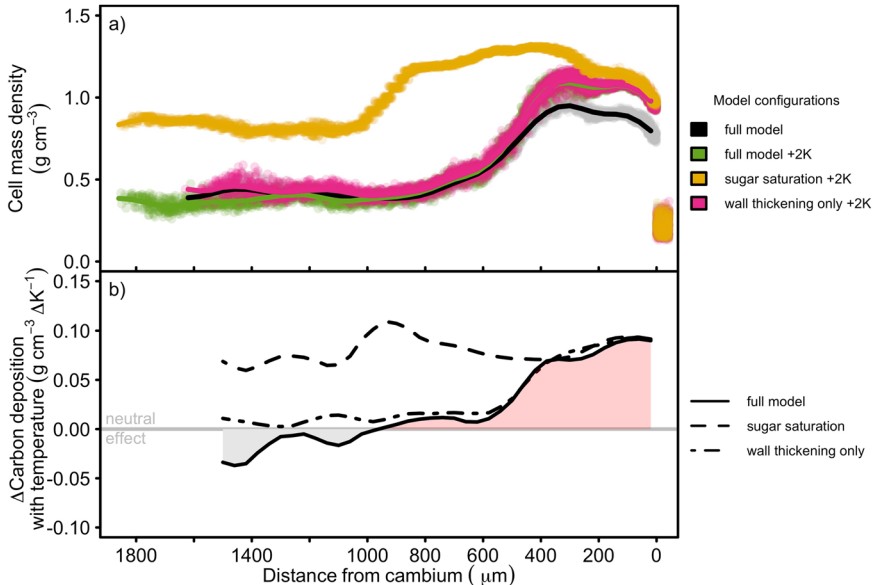

**Fig. 3 | Effect of temperature on simulated cell-mass density across the radial profile under different model configurations and forcings in *Pinus sylvestris* L. at the same site as in Fig. 2, 1995. a** Simulated cell-mass density using observed temperatures ("full model") and with +2 K applied daily to both cell enlargement and wall thickening ("full model +2 K")' or only wall thickening ("wall thickening only +2 K"). Also shown is the result of applying the temperature increment to both cell growth and wall thickening but with the carbohydrate limitation to cell-wall growth removed ("sugar saturation +2 K"), as in Fig 2b. Each circle is an individual cell at the end of the year, and solid lines are mean density profiles across 100 radial files for each model configuration (running means over three 40 μm-wide bins). **b** Ratio of change in density to change in temperature calculated by subtracting the mean profiles of 100 radial files at +2 K from the mean profiles at observed temperatures ("full model"). The dot-dash line is the difference between a run where +2 K was applied to wall thickening only and a run using observed temperatures ("wall thickening only"). Also shown (dashed line) is the difference between a run with both non-limiting carbohydrates and +2 K applied to cell growth and wall thickening and one when carbohydrates are non-limiting but at observed temperatures ("sugar saturation").

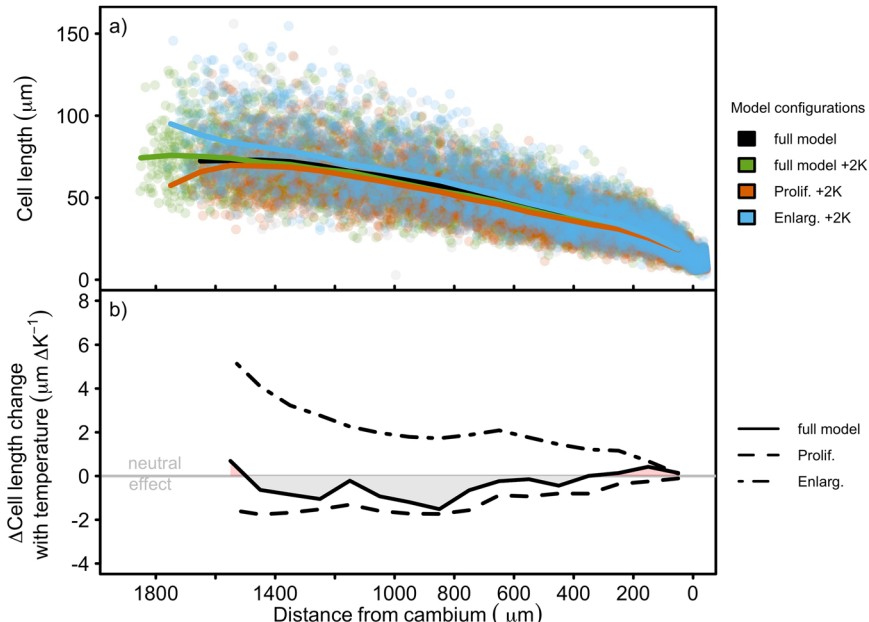

**Fig. 4 | Effect of temperature on simulated cell radial length across the final ring under different model forcings in *Pinus sylvestris* L. growing at the same site as in Fig. 2, 1995. a** Simulated cell radial lengths at observed ("full model") and observed +2 K ("full model +2 K") temperatures with the full model. Also shown are results with the temperature increment of +2 K applied to only proliferating cells ("Prolif. +2 K"), or only enlargement-zone cells ("Enlarg. +2 K"). Each point is an individual cell at the end of the year and solid lines are mean cell-length profiles using a running mean over three adjacent 100 μm-wide bins. 100 radial files were simulated for each forcing. **b** Radial profiles of ratios of change in cell length to change in temperature using results in **a**), created by computing the difference in cell length profile from a full model run with observed temperatures from a run with an increase of +2 K each day applied to the full model, or the difference between a full model run at observed temperatures and a run with +2 K applied to proliferating cells only ("Prolif."), and the difference between a full model run at observed temperatures and one with a temperature effect on enlargement-zone cells only ("Enlarge.").

determining final cell sizes were determined by comparing simulations in which temperature changes were applied to either the proliferation or enlargement-only zones alone, or to both. These simulations show that the compensatory mechanism is a subtle one. Higher temperatures in the enlargement-only zone alone indeed increase cell size, but higher temperatures in the proliferation zone alone decrease it (Fig. 4b). Higher rates of proliferation with increased temperature increase the rate at which cells are added to the enlargement-only zone (an average over the growing season of ca. 3 additional cells/file, +7%, for a 2 K rise), rapidly pushing the zones outwards and so causing the duration of enlargement to decline (this can be envisaged in Supplementary Fig. 2 as a steepening of the lines joining cells across the enlargement-only zone), compensating for the increased rate of enlargement.

### Overall effects of temperature

To determine the effects of temperature on multiple tree ring features, a series of simulations was performed with the full model as above, but with step changes in temperature added to the observed daily temperatures and applied to both the enlargement and wall thickening rates (Fig. 5). Ring width increases strongly and linearly with temperature across the entire range sampled. In contrast, total ring mass increases more strongly than ring width at negative temperature anomalies, but for higher temperatures it saturates, with little change > +5 K. As was shown in Fig. 3, the increase in ring width with temperature is due to an increase in the width of earlywood. This causes mean-ring density to decrease for positive temperature increments (Fig. 5; N.B. density decreases have been observed in different species over recent decades[35]). The increase in ring width is mainly due to an increase in the number of cells, rather than their size, which is consistent with the findings above of opposing effects of proliferation and enlargement rates on final cell sizes (Fig. 4). Removing the carbohydrate constraint on wall thickening causes total mass to increase very strongly with temperature as earlywood wall thickening is no longer carbon-limited because of its distance from the phloem (this

constraint increases with temperature due to greater cell-phloem distances; cf. Supplementary Fig. 2). Maximum density increases at the same rate as total width at negative temperature anomalies, but gradually saturates as temperature increases.

The diverging responses of width and mass to temperature have significant implications for the measurement of tree mass growth trends and hence carbon uptake, which is usually inferred from volume increments[1]. In addition, the saturation of maximum density with temperature has implications for its use as a proxy for past climate, and may well play a role in the "divergence problem"[37].

## Discussion

The mechanisms explored here suggest that conifer ring anatomical structures largely arise from the interactions of the activities and spatial configurations of seaonally-evolving developmental zones with the distribution of carbohydrates across the radial file. These interacting factors buffer the response of features such as cell sizes, the earlywood-latewood transition, and earlywood density to temperature variation. Crucially, the diffusion of carbohydrates from the phloem plays a major role in the distribution of final mass across the mature tree ring and its response to temperature. These findings have implications for our understanding of the response of carbon sequestration in trees to changing temperature and carbohydrate supply, such as may be caused by rising atmospheric $CO_2$. Moreover, the framework presented here provides a mechanistic basis for the explicit treatment of tree growth in global vegetation models, facilitating a balanced source/sink approach as advocated in a growing body of literature[3,38–40].

We hypothesise that factors such as hydrological and nutrient constraints could be mechanistically incorporated into our framework through effects on the rate of cell enlargement. In addition, the dynamics of carbon supply from the phloem can be simulated through a treatment of the dynamics of labile sugars (Friend et al.[3] provide a possible approach for this), and extension to other species, including non-conifers, will be the focus of future work (e.g. by incorporating approaches developed by Drew et al.[17]). Reformulation of global vegetation models based on the approach described here has the potential to reconcile inconsistencies between current models and observations (e.g. in relation to increasing $CO_2$), with significant consequences for predictions of tree carbon storage, and hence the global carbon cycle.

## Methods

### Overall framework

Cells in our model are arranged along independent radial files, with each cell in one of either the proliferation, enlargement-only, wall thickening, or mature zones, depending on the distance of the cell's centre from the inside edge of the phloem and the time of year. Only cells that contribute to the formation of xylem tracheids are treated explicitly. A daily timestep is used, on which cells in the proliferation and enlargement-only zones can enlarge in the radial direction if these zones are non-dormant, and on which secondary-wall thickening can occur in the wall thickening zone. Cells in the proliferation zone divide periclinally if they reach a threshold radial length. Cell-size control at division is intermediate between a critical size and a critical increment[22]. Mother cells divide asymmetrically, with the subsequent relative growth rates of the daughters inversely proportional to their relative sizes. Size at division and asymmetry of division are computed with added statistical noise[22], and therefore the model is run for an ensemble of independent radial files with perturbed initial conditions.

### Equations and parameters

**Cell enlargement and division.** Cells in the proliferation and enlargement-only zones, when not dormant, enlarge in the radial

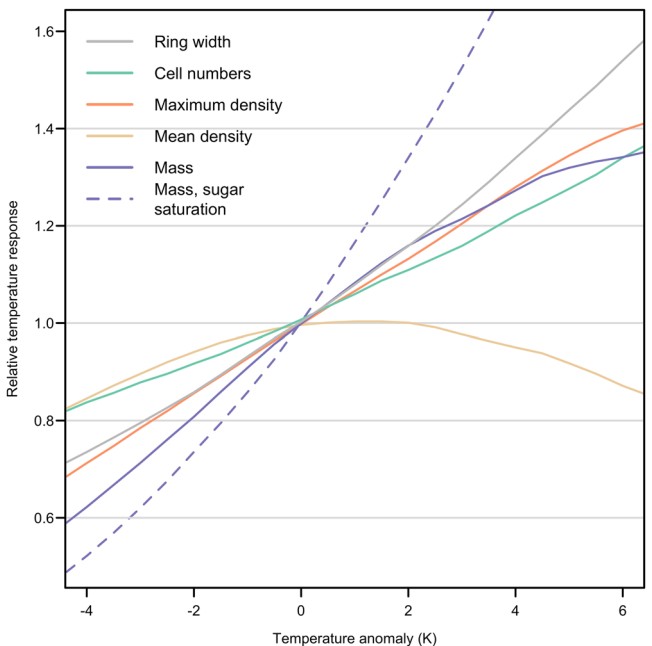

**Fig. 5 | Temperature effects on simulated growth in *Pinus sylvestris* L. at the site used in Fig. 2, 1995.** Each line is the running mean for 100 radial files of the variable's response over a 1.5 K-wide window (i.e. across three 0.5 K increments). Temperature anomaly applied as a step increment to the observed daily values used to compute cell enlargement and wall thickening.

**Table 1 | Model parameters calibrated to observations**

| Symbol | Description | Equation | Value | Units | Source |
|---|---|---|---|---|---|
| $\eta$ | Resistance to carbohydrate diffusion between cells | (15), (16) | 4360 | day ml$^{-1}$ | [23] |
| $\mu_0$ | Relative radial growth rate of cell at $T_0$ | (1) | 0.0570 | $\mu$m $\mu$m$^{-1}$ day$^{-1}$ | [23] |
| $\omega$ | Normalised rate of cell-wall mass growth | (9) | 19.4 | mg ml$^{-1}$ day$^{-1}$ | [23] |
| $E_a$ | Effective activation energy for cell enlargement | (1) | 0.374 | eV | [31] |
| $E_{aw}$ | Effective activation energy for wall building | (9) | 1.43 | eV | [31] |
| $K_m$ | Effective Michaelis constant for wall building | (8) | 5.1 | mg ml$^{-1}$ | [23] |

Calibration of $\eta$, $\omega$, and $K_m$ used predicted carbohydrate concentrations in cells 1, 13, and 22 and cell masses in cells 10, 13, and 22 (counting from the initial), $\mu_0$ used the final ring width, $E_a$ used the ring width response to temperature, and $E_{aw}$ used the maximum density response to temperature.

**Table 2 | Parameters used in the model that are taken directly from literature**

| Symbol | Description | Equation | Value | Units | Source |
|---|---|---|---|---|---|
| $f$ | Mode of cell size regulation | (7) | 0.48 | - | [22] |
| $g_{asym}$ | Size growth dependence on division asymmetry | (3) | 0.43 | - | [22] |
| $L_a$ | Cell axial length | (11), (14) | 2680 | $\mu$m | [45] |
| $\rho$ | Cell-wall mass density | (12), (13) | $1.54 \times 10^3$ | mg[DM] ml$^{-1}$ | [46] |
| $\sigma_a$ | Standard deviation of division asymmetry | (6) | 0.105 | - | [22] |
| $\sigma$ | Standard deviation of birth size | (7) | 0.227 | - | [22] |

direction at a rate dependent on temperature and relative sibling birth size. A Boltzmann-Arrhenius approach is used for the temperature dependence[30]:

$$\mu = \mu_0 e^{\frac{E_a}{k}\left(\frac{1}{T_0} - \frac{1}{T}\right)} \tag{1}$$

where $\mu$ is the relative rate of radial cell growth at temperature $T$ ($\mu$m $\mu$m$^{-1}$ day$^{-1}$), $\mu_0$ is $\mu$ at temperature $T_0$ (=283.15 K), $E_a$ is the effective activation energy for cell enlargement, $k$ is the Boltzmann constant (i.e. $8.617 \times 10^{-5}$ eV K$^{-1}$), and $T$ is temperature (K). $\mu_0$ was calibrated to an observed mean radial file length at the end of the elongation period dataset[23] (Table 1; see "Observations"), and $E_a$ was calibrated to an observed temperature dependence of annual ring width dataset[31] (Table 1; Supplementary Fig. 4; see "Observations").

Radial length of an individual cell then increases according to:

$$\Delta L_r = L_r(e^{\epsilon\mu} - 1) \tag{2}$$

where $\Delta L_r$ is the radial increment of the cell ($\mu$m day$^{-1}$), $L_r$ is the radial length of the cell ($\mu$m), and $\epsilon$ is the cell's growth dependence on relative birth size, given by[22]:

$$\epsilon = 1 - g_{asym}\alpha_b \tag{3}$$

where $g_{asym}$ is the strength of the dependence of relative growth rate on asymmetric division (Table 2; unitless), and $\alpha_b$ is the degree of asymmetry relative to the cell's sister[22] (scalar):

$$\alpha_b = \frac{L_{r,b} - L_{r,b}^{sis}}{L_{r,b} + L_{r,b}^{sis}} \tag{4}$$

where $L_{r,b}$ is the radial length of the cell at birth ($\mu$m) and $L_{r,b}^{sis}$ is the radial length of its sister at birth ($\mu$m), which are calculated stochastically[22]:

$$L_{r,b} = L_{r,d}(0.5 - Z_a) \tag{5}$$

$$L_{r,b}^{sis} = L_{r,d}(0.5 + Z_a) \tag{6}$$

where $L_{r,d}$ is the length of the mother cell when it divides ($\mu$m) and $Z_a$ is Gaussian noise with mean zero and standard deviation $\sigma_a$ (Table 2; $-0.49 \le Z_a \le 0.49$ for numerical stability).

Length at division is derived as[22]:

$$L_{r,d} = fL_{r,b} + \chi_b(2 - f + Z) \tag{7}$$

where $f$ is the mode of cell-size regulation (Table 2; unitless), $\chi_b$ is the mean cell birth size (Table 3; $\mu$m), and $Z$ is Gaussian noise with mean zero and standard deviation $\sigma$ (Table 2).

The first cell in each radial file is an initial, which produces phloem mother cells outwards and xylem mother cells inwards. It grows and divides as other cells in the proliferation zone, but on division one of the daughters is stochastically assigned to phloem or xylem, the other remaining as the initial. The probability of the daughter being on the phloem side is $f_{phloem}$ (Table 3).

**Cell-wall growth.** Both primary and secondary cell-wall growth are influenced by temperature, carbohydrate concentration, and lumen volume. A Michaelis-Menten equation is used to relate the rate of wall growth to the concentration of carbohydrates in the cytoplasm:

$$\Delta M = \frac{\Delta M_{max}\theta}{\theta + K_m} \tag{8}$$

where $\Delta M$ is the rate of cell-wall growth (mg cell$^{-1}$ day$^{-1}$), $\Delta M_{max}$ is the carbohydrate-saturated rate of wall growth (mg cell$^{-1}$ day$^{-1}$), $\theta$ is the concentration of carbohydrates in the cell's cytoplasm (mg ml$^{-1}$), and $K_m$ is the effective Michaelis constant (mg ml$^{-1}$; Table 1).

The maximum rate of cell-wall growth, $\Delta M_{max}$, is assumed to depend linearly on lumen volume (a proxy for the amount of machinery for wall growth), and on temperature as in Eq. (1):

$$\Delta M_{max} = \omega V_l e^{\frac{E_{aw}}{k}\left(\frac{1}{T_0} - \frac{1}{T}\right)} \tag{9}$$

where $\omega$ is the normalised rate of cell-wall mass growth (i.e. the rate at $T_0$; Table 1; mg ml$^{-1}$ day$^{-1}$), $V_l$ is the cell lumen volume (ml cell$^{-1}$), and $E_{aw}$ is the effective activation energy for wall building (eV; Table 1). $\omega$ and $K_m$ were calibrated to an observed distribution of carbohydrates[23]

**Table 3 | Parameters used in the model that are calculated from observations**

| Symbol | Description | Equation | Value | Units | Source |
|---|---|---|---|---|---|
| $a_p$ | Proliferation zone intercept | (17) | −50.9 | $\mu m$ | [23] |
| $a_e$ | Enlargement-only zone intercept | (17) | DOY≥185: −744 DOY<185: 1257 | $\mu m$ | [23] |
| $a_t$ | Wall-thickening zone intercept | (17) | −427 | $\mu m$ | [23] |
| $b_p$ | Proliferation zone slope | (17) | 7.47 | $\mu m\ hr^{-1}$ | [23] |
| $b_e$ | Enlargement-only zone slope | (17) | DOY≥185: 51.7 DOY<185: −5.01 | $\mu m\ hr^{-1}$ | [23] |
| $b_t$ | Wall-thickening zone slope | (17) | 68.8 | $\mu m\ hr^{-1}$ | [23] |
| $f_{phloem}$ | Probability of initial dividing to phloem mother | - | 0.315 | - | [28] |
| $L_t$ | Cell tangential length | (11), (14) | 50.1 | $\mu m$ | [23] |
| $W_p$ | Primary cell-wall thickness | (14) | 0.8 | $\mu m$ | [23] |
| $\chi_b$ | Mean cell radial length at birth | (7) | 8.8 | $\mu m$ | [23] |
| $\theta_p$ | Phloem carbohydrate concentration | (16) | 95 | $mg\ ml^{-1}$ | [23] |

(see next section). $E_{aw}$ was calibrated to an observed temperature dependence of maximum density[31] (Table 1; see "Observations").

Lumen volume is given by:

$$V_l = V_c - V_w \qquad (10)$$

where $V_c$ is total cell volume (ml cell$^{-1}$) and $V_w$ is total wall volume (ml cell$^{-1}$). Cells are assumed cuboid and therefore $V_c$ is given by:

$$V_c = L_a L_t L_r / 10^{12} \qquad (11)$$

where $L_a$ is axial length ($\mu m$; Table 2) and $L_t$ is tangential length ($\mu m$; Table 3). $V_w$ is given by:

$$V_w = M/\rho \qquad (12)$$

where $M$ is wall mass (mg cell$^{-1}$) and $\rho$ is wall-mass density (Table 2; mg[DM] ml$^{-1}$).

Cells in the proliferation and enlargement-only zones only have primary cell walls. $\Delta M_{max}$ (Eq. (9)) is therefore given the following limit:

$$\Delta M_{max} = \min(\Delta M_{max}, \rho V_{w_p} - M) \qquad (13)$$

where $V_{w_p}$ is the required primary wall volume:

$$V_{w_p} = V_c - (L_a - 2W_p)(L_t - 2W_p)(L_r - 2W_p)/10^{12} \qquad (14)$$

where $W_p$ is primary cell-wall thickness (Table 3; $\mu m$).

**Carbohydrate distribution.** The distribution of carbohydrates across each radial file is calculated independently from the balance of diffusion from the phloem and the uptake into primary and secondary cell walls. The carbohydrate concentration in the phloem is prescribed at the mean value observed across the three observational dates in[23], as described below in "Observations", and the resulting concentration in the cytoplasm of the furthest living cell from the phloem is solved numerically. The inside wall of this cell is assumed to be impermeable to carbohydrates and therefore provides the inner boundary to the solution. It is assumed that the rate of diffusion across each file is rapid relative to the rate of cell-wall building, and therefore concentrations are assumed to be in equilibrium on each day. Carbohydrate diffusion between living cells is assumed to be proportional to the concentration gradient:

$$q_i = (\theta_{i-1} - \theta_i)/\eta \qquad (15)$$

where $q_i$ is the rate of carbohydrate diffusion from cell $i-1$ to cell $i$ (mg day$^{-1}$) and $\eta$ is the resistance to flow between cells (calibrated to the observed distribution of carbohydrates[23], see next section; Table 1; day ml$^{-1}$).

As it is assumed that carbohydrates cannot diffuse between radial files, at equilibrium the flux into a given cell must equal the rate of wall growth in that cell plus the wall growth in all cells further along the radial file away from the phloem. From this it can be shown that the equilibrium carbohydrate concentration in the furthest living cell from the phloem in each radial file is given by:

$$\theta_n = \theta_p - \eta \sum_{i=1}^{n} \sum_{j=i}^{n} \Delta M_j \qquad (16)$$

where $\theta_p$ is the concentration of carbohydrates in the phloem (Table 3; mg ml$^{-1}$) and $n$ is the number of living cells in the file (phloem mother cells are ignored for simplicity). The rate of wall growth in each cell depends on the concentration of carbohydrates (Eq. (8)), and therefore $\theta_n$ must be found that results in an equilibrium flow across the radial file. This is done using Brent's method[41] as implemented in the "ZBRENT" function[42].

**Zone widths.** The widths of the proliferation, enlargement-only, and secondary wall thickening zones vary through the year, and are fitted to observations on three dates[23] (see Supplementary Fig. 2 and "Observations"). Linear responses to daylength were found, which are therefore used to determine widths for the observational period and later days:

$$z_k = a_k + b_k dl; DOY \geq 185 \qquad (17)$$

where $z_k$ is the distance of the inner edge of the zone from the inner edge of the phloem ($\mu m$), $k$ is proliferation ($p$), secondary wall thickening ($t$), or enlargement-only ($e$), $a_k$ and $b_k$ are constants (Table 3), dl is daylength (s), and DOY is day-of-year. The proliferation zone width on earlier days when non-dormant was fixed at its DOY 185 width (assuming this to be its maximum, and that it would reach its maximum very soon after cambial dormancy is broken in the spring). During dormancy, the proliferation zone width is fixed at its value on DOY 231 (the first day of dormancy[23]). The enlargement-only zone width prior to DOY 185, the first observational day, is assumed to be a linear extension of the rate of change after DOY 185. The wall-thickening zone width plays little role prior to DOY 185 at the focal site, and so was set to its Eq. (17) value each earlier day. On all days the condition $z_t \geq z_e \geq z_p$ is imposed, and zone widths cannot exceed their values at 24 h daylength (necessary for sites north of the Arctic circle). Supplementary Figure 2 shows the resulting progression of zone widths through the year, together with the observed values.

**Dormancy**. Proliferation was observed to be finished by DOY 231[23], and so the proliferation and enlargement-only zones are assumed to enter dormancy then. Release from dormancy in the spring is calculated using an empirical thermal time/chilling model[33]. It was necessary to adjust the asymptote and temperature threshold of the published model because the heat sum on the day of release calculated from observations in Sweden (see "Observations") was much lower than reported for Sitka spruce buds in Britain in the original work:

$$\mathrm{dd}_{req} = 15 + 4401.8 e^{-0.042\mathrm{cd}} \tag{18}$$

where $\mathrm{dd}_{req}$ is the required sum of degree-days (°C) from DOY 32 for dormancy release and cd is the chill-day sum from DOY 306. The degree-day sum is the sum of daily mean temperatures above 0 °C, and the chill-day sum is the number of days with mean temperatures below 0 °C. Dormancy can only be released during the first half of the year.

### Simulation protocols
Each simulation consisted of an ensemble of 100 independent radial files. Each radial file was initialised by producing a file of 100 cells with radial lengths $\chi_b(1+Z_a)$, allowing these to divide once, ignoring the second daughter from each division, and then limiting the remaining daughters to only those falling inside the proliferation zone on DOY 1. Values for $\epsilon$ (the relative growth of daughter cells) and $L_{r,d}$ (the radial length at division) were derived for each cell. The main simulations were conducted at the observation site in boreal Sweden (64.35°N, 19.77°E) over 1951–1995 to capture the growth period of the observed trees[23]. Results are mostly presented for 1995 when the observations were made. Simulations for calibration of the effective activation energies (i.e. $E_a$ and $E_{aw}$) were performed at 68.26°N, 19.63°E in Arctic Sweden over 1901–2004[31]. Daily mean temperatures for both sites were derived from the appropriate gridbox in a 6 h 1/2 degree global-gridded dataset[43].

### Observations
Observations of cellular characteristics and carbohydrate concentrations[23] were used to derive a number of model parameters, and to test model output (model calibration and testing were performed using different outputs). According to the published study we used, samples were cut from three 44 yr old Scots pine trees growing in Sweden (64°21′ N; 19°46′ E) at different times during the growing season. 30 $\mu$m thick longitudinal tangential sections of the cambial region were made, and the radial distributions of soluble carbohydrates measured using microanalytical techniques[23]. Cell sizes, wall thicknesses, and positions in their Fig. 1[23], an image of transverse sections on three sampling dates, were digitised using "WebPlotDigitizer"[44]. These, together with the numbers of cells in each zone and their sizes given in the text of that paper, were used to estimate zone widths, which were then regressed against daylength to give the parameters for Eq. (17) (Table 3), mean cell size in the proliferation zone on the first sampling date (used to derive $\chi_b$; Table 3), mean cell tangential length (Table 3), and final ring width (used to calibrate $\mu_0$; Table 1). The thickness of the primary cell wall (Table 3) was derived by plotting cell-wall thickness against time and taking the low asymptote.

The distributions of carbohydrates along the radial files on the last sampling date for "Tree 1" and "Tree 3" (results for "Tree 2" were not shown for this date) shown in Fig. 2 of the observational paper[23] were calculated. The masses for each of sucrose, glucose, and fructose in each 30 $\mu$m section were digitised using the same method as for cell properties and then summed and converted to concentrations, with the results shown in Supplementary Figure 5. Mean observed carbohydrate concentrations and cell masses at four points were used to calibrate values for the $\eta$, $\omega$, and $K_m$ parameters in Table 1. Calibration

was performed by minimising the summed relative error across the observations.

The calibration target for the effective activation energy for wall deposition (i.e. $E_{aw}$) was the observed relationship between maximum density and mean June-July-August temperature over 1901-2004 in northern Sweden[31] (Supplementary Fig. 3), and for the effective activation energy for cell enlargement the relationship between ring width and temperature (i.e. $E_a$) target was the same study (Supplementary Fig. 4). These observations were made on living and subfossil Scots pine sample material from the Lake Torneträsk area (68.21–68.31°N; 19.45–19.80°E; 350–450 m a.s.l.) using X-ray densitometry for maximum density, and standardised to remove non-climatic information[31].

### Reporting summary
Further information on research design is available in the Nature Portfolio Reporting Summary linked to this article.

## Data availability
The model output data generated in this study and visualisation scripts can be accessed at: https://doi.org/10.5281/zenodo.7441946 Data used for calibration of the temperature sensitivity of primary and secondary wall growth is from[31] and data for the calibration of the local availability of carbohydrates are from[23]; The remaining parameters were derived from multiple studies (see Table 2 and Table 3, column "Source").

## Code availability
The model code used in this publication is available at https://doi.org/10.5281/zenodo.7357545 Model analysis scripts in this publication can be found at the link provided in the Data availability statement, above.

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

## Acknowledgements

This work has benefitted greatly from discussions with numerous people, in particular Flurin Babst, Soumaya Belmecheri, Ulf Büntgen, Henri Cuny, Patrick Fonti, David Frank, Andrew Hacket Pain, Christian Körner, Ben Poulter, Tim Tito Rademacher, Cyrille Rathgeber, Andrew Richardson, and Valerie Trouet. A.D.F. acknowledges support from the Natural Environment Research Council under grant no. NE/W000199/1. Q.T. wishes to thank the Cambridge Faculty of Mathematics for support via the CMP scheme. A.H.E-S. acknowledges support from the European Research Council under the European Union Horizon 2020 Programme (grant no. 758873, TreeMort). This study contributes to the Strategic Research Areas BECC and MERGE. For the purpose of open access, the author has applied a Creative Commons Attribution (CC BY) licence to any Author Accepted Manuscript version arising from this submission.

## Author contributions

A.D.F. conceived the underlying theoretical framework, determined the observed quantities from the literature, wrote and analysed the model,

and drafted the manuscript. A.H.E-S. provided feedback on conceptual ideas and input to the writing of the manuscript, and drew most of the figures. Q.T. provided advice on the implementation of the model's mathematical constructions, verified their formulation, and helped to edit the manuscript.

## Competing interests

The authors declare no competing interests.
