## [Peer Review File · Nature Communications]

Wood structure explained by complex spatial source-sink interactionsReviewer #1 (Remarks to the Author):

General comments:

This contribution presents a mechanistic model of xylem cell production, enlargement and maturation. Those are the basic processes of "wood formation" that is both crucial and coarsely simulated to date in terrestrial ecosystem models (TEMs). Such models are rare in the literature (but see Vaganov et al. 2006; Cabon et al. 2020). This model (as the others) has the potential to become a module of TEMs.

Another merit is the model formalism, that is both new and interesting. It poses that sucrose fueling from the phloem and temperature are key in the production and maturation of xylem cells. The author does a very nice job in dissecting (through sensitivity analysis and switching on/off the processes modelled) the influences of both drivers on different aspects of wood formation. One main conclusion is that sucrose fueling is one key driver of wood formation and explains the differential density profile (lighter earlywood, denser latewood) usually observed in coniferous trees (but also in angiosperms).

So overall I'm positive about this paper.

However, I think the paper does not do justice to previously existing models (cited above). The formalism and objectives are different indeed, but I would like to read a paragraph / section into which the author informs the reader about such differences. Explaining to what extent is his model new and able to shedding new light on wood formation processes.

Another aspect of concern to me is that the author chose to focus on carbon fueling and temperatures as determinants of wood formation. However, water potential probably plays a large role, as proposed by the classical formulation of Lockhart (1965) about cell expansion. It was recently considered by Cabon et al. (2020) who based a model on it, and predicted with good accuracy the formation of wood tracheid along a large environmental gradient in two conifer species (*Picea abies* and *Larix decidua*). I understand that the author of the current contribution based on data from the northern part of the *Pinus sylvestris* distribution zone to calibrate and verify his model. However, this questions the genericity of his approach. Discussion about these aspects also is required: (i) transferability of the results to other sites given the climate specificity of the observation data onto which the model is built; and (ii) absence of consideration of water potential in wood formation processes.

The author claims (L203-206) that his model could serve as a basis for representing wood formation in TEMs. This is desirable indeed, but considering that TEMs do not include explicit representations of phloem fueling, I would also expect the author to give leads about how to adapt and implement his model in TEMs.

Specific comments:

L77-78: recently Delpierre et al. (2019) published parameterizations for such models on various conifers, among which *Pinus sylvestris*.

L81: remove "highly". "Are plausible" is enough.

Figure 1 caption (and other figures using running means): I'm unsure what "0.06-fraction running mean over 0.02-fraction bins." Mean. Please rephrase.

Figure 2 inset: Unsure i understand the y-scale unit. It does not correspond with the magnitude of change observed on the main graph of Fig. 2 (e.g. on the inset, the sensitivity under the condition "saturating carbohydrates" is about 0.10 g/cm³/K, when on the main figure we switch, e.g. at 750 μm from the cambium from 0.5 g/cm³ (default

condition) to 1.1 g/cm³ (saturating CH condition) for a +2K increase: I expect a sensitivity of $(1.1-0.5)/2 = 0.30 \text{ g/cm}^3/\text{K}$.

L157: "shown"

L206: another paper of interest is Guillemot et al. (2017)

Table 3: How was the DoY 185 limit set? Arbitrarily? Summer solstice is on DoY 172, so one cannot argue that this is a "natural" limit.

Equations 11 and 14: you have to multiply (not divide) by 10^{12} to convert $\mu\text{m}^3 (= (10^{-6})^3 = 10^{-18} \text{ m}^3)$ to mL ($= 10^{-6} \text{ m}^3$)

L272: "the phloem carbohydrate is prescribed": how is it prescribed? Constant along the season? Varying? Based on what data?

Equation 17: what is the applicability range of this parameterization? whole distribution area of *Pinus sylvestris*? Site-specific?

L296: when you refer to dormancy here, i assume this is dormancy of the cambium (i.e. no more cell division in the proliferation zone). This is unclear.

L351: "Wood formation in trees" (not "gormation")

References cited:

Cabon, A., Peters, R. L., Fonti, P., Martínez-Vilalta, J., & De Cáceres, M. (2020). Temperature and water potential co-limit stem cambial activity along a steep elevational gradient. *New Phytologist*, 226(5), 1325-1340.

Delpierre, N., Lireux, S., Hartig, F., Camarero, J. J., Cheaib, A., Čufar, K., ... & Huang, J. G. (2019). Chilling and forcing temperatures interact to predict the onset of wood formation in Northern Hemisphere conifers. *Global change biology*, 25(3), 1089-1105.

Guillemot, J., Francois, C., Hmimina, G., Dufrêne, E., Martin-StPaul, N. K., Soudani, K., ... & Delpierre, N. (2017). Environmental control of carbon allocation matters for modelling forest growth. *New Phytologist*, 214(1), 180-193.

Lockhart, J. A. (1965). An analysis of irreversible plant cell elongation. *Journal of theoretical biology*, 8(2), 264-275.

Vaganov, E. A., Hughes, M. K., & Shashkin, A. V. (2006). Growth dynamics of conifer tree rings: images of past and future environments (Vol. 183). Springer Science & Business Media.

Reviewer #2 (Remarks to the Author):

Revision and comments on the manuscript

"Wood structure explained by complex spatial source-sink interactions" by Andrew D. Friend

The main problem this manuscript address is the lack of understanding of how wood structure in Scots pine is formed. The research uses a sophisticated model to estimate wood structure characteristics and how and why it predicts the formation and transition from Earlywood to latewood. One of the main conclusions highlighted is that the distribution of carbohydrates available influences these wood density variations during

xylogenesis, which is very interesting. However, I could not understand from the manuscript how the model operates, besides reading through the 12 assumptions listed in the text. I think this paper is not acceptable in its current stage because this kind of sophisticated modeling should be much better explained than in this type of short format.

From what I understand from the readings, this manuscript concludes that the wood formation of conifer spp, "now generalized," depends on the interaction of many factors, which in my opinion, is not scientifically satisfying, at least how is presented now. This statement might sound subjective, but I try to explain in the following paragraphs with the best of my intentions. I think it all comes down to the focus and structure of the manuscript that leaves me with this idea.

The focus of the manuscript and claims are based on the results are depending heavily on the model output. The model has a high level of parametrization and an extensive list of assumptions. I am not saying this is inherently bad, but this should be much better explained, in detail, so there are no loose ends, or at least those loose ends are well addressed.

One of the claims in the article, states that this model can help improve the carbon cycle models is a bit of a long shot because it would be impossible or at least not reasonable to extrapolate this model to all tree spp. and climates of the world. At least this is how it looks form the narrative. Also, it is not explicit how this approach can help the land surface models that the author refers to? One of the concerns I have, is how the author can justify those claims when knowing that there are considerable variations in xylogenesis and wood structure across the world? In the text of the manuscript is not explicitly stated how this model is extrapolable to other species.

Another aspect that I find weak is that the methods in this manuscript are set on a lower profile (less critical) when I think this should be a powerful factor on this type of manuscript. I think that the details on the description of the "observations" should play a much bigger role in the paper's main body and explain with much more details and clearer language.

The results and figures are poorly presented, and the figures could be more carefully produced, with higher quality, so it is easy for the reader to follow the main messages. Additionally, the narrative of the paper is a bit complicated to follow. Many important concepts in the manuscript are not well introduced, for example; 1) The developmental stages, 2) The observations made on *Pinus silvestris* we are left with only the fact the author made "critical observations" that is all until we arrive in the figures with plenty lines showing the "observations". 4) In the last sentence before the conclusions, there is a statement with the "divergence problem" which has not been properly introduced or addressed throughout the manuscript.

Reviewer #3 (Remarks to the Author):

The manuscript "Wood structure explained by complex spatial source-sink interactions" presents a model of cell growth in cambium to form a framwork to explain possible mechnisms to form wood of a tree so as to attribute tree ring density profile. The well known theory, oberavation and suitable assumption are employed to make simulation test. And significant fresh and new ideas are addressed: NSC diffusion and its responses to temprature is a dominant reason to determinate the tree ring density (hence wood volume mass content).From this idea, the pressent time' volum based tree carbon storage computing should be updated by considering temperature effect on tree ring density (via earlywood and latewood). Therefore this paper will be forcused on by many researchers even common readers. I think this paper should be accepted to publish.

suggestions:

The concept and structure of this model could be clearer if there is a figure to present the whole framework.

This model does not simulate the NSC dynamics (but uses the observed NSC). Recently, some studies have constructed the physiological models that modeled the NSC dynamics and cell growth, e.g. the CASSIA model (Schiestl-Aalto et al. (2015)) and FORCCHN2 model (Fang et al. (2020)). these references's findings and ideas may strengthen the understanding of this work.

Specific comments:

Fig. 1: related questions.

Sweden site modeled data were comparison to other 4 sites's observation, which may be questioned. Because this paper regarded climate's effects (temperature) as important factors, those other 4 sites climate condition need to be compared to Sweden site.

My understanding is that except for the saturating experiment run, the carbohydrates always keep as observation. If so, it seems the single ring mass integration should be almost equal in different experiment runs. However from figure, it seems that fixed zone, fixed temperature, no dormancy produce very different mass amount in a same tree ring.

Fig. 2 related questions.

+2K experiment runs may increase ring width (from fig) ?

Only wall thickening experiment run seem to allocate more mass in tree ring, however carbohydrate amount should be fixed to same value if my understanding is right.

Table 1: μ_0 is the final ring width? In this work, the μ_0 is a relative value, and it is equal to $0.0559 \mu\text{m} \mu\text{m}^{-1}\text{day}^{-1}$. E_a is the ring width response to temperature? The E_a may represent the density response to temperature.

Line 126-127: the density distribution was determined by the activities of cambium via NSC. beside the temperature' effect, hydraulics from xylem also influence the density distribution.

Line 323: The observations from the published paper, and you could add a simple description of the observations.

Responses to the reviewers' comments on the paper by Friend, AD: "Wood structure explained by complex spatial source-sink interactions"

We (two additional authors have been added who have helped with the resubmission) are very grateful for the attention the reviewers have taken in assessing our paper and we provide responses to their comments below. We have put the reviewers' comments in *italic* and our responses in **bold** in order to distinguish them from each other.

Reviewer #1 (Remarks to the Author):

General comments:

This contribution presents a mechanistic model of xylem cell production, enlargement and maturation. Those are the basic processes of "wood formation" that is both crucial and coarsely simulated to date in terrestrial ecosystem models (TEMs). Such models are rare in the literature (but see Vaganov et al. 2006; Cabon et al. 2020). This model (as the others) has the potential to become a module of TEMs.

Thank you for recognising the potential of our model.

Another merit is the model formalism, that is both new and interesting. It poses that sucrose fueling from the phloem and temperature are key in the production and maturation of xylem cells. The author does a very nice job in dissecting (through sensitivity analysis and switching on/off the processes modelled) the influences of both drivers on different aspects of wood formation. One main conclusion is that sucrose fueling is one key driver of wood formation and explains the differential density profile (lighter earlywood, denser latewood) usually observed in coniferous trees (but also in angiosperms).

So overall I'm positive about this paper.

However, I think the paper does not do justice to previously existing models (cited above). The formalism and objectives are different indeed, but I would like to read a paragraph / section into which the author informs the reader about such differences. Explaining to what extent is his model new and able to shedding new light on wood formation processes.

Thank you for this suggestion. In response we have added a paragraph briefly discussing the range of existing approaches and how/why our model is different. This is on lines 43–72 of the revised manuscript.

*Another aspect of concern to me is that the author chose to focus on carbon fueling and temperatures as determinants of wood formation. However, water potential probably plays a large role, as proposed by the classical formulation of Lockhart (1965) about cell expansion. It was recently considered by Cabon et al. (2020) who based a model on it, and predicted with good accuracy the formation of wood tracheid along a large environmental gradient in two conifer species (*Picea abies* and *Larix decidua*).*

*I understand that the author of the current contribution based on data from the northern part of the *Pinus sylvestris* distribution zone to calibrate and verify his model. However, this questions the genericity of his approach. Discussion about these aspects also is required: (i)*

transferability of the results to other sites given the climate specificity of the 11 observation data onto which the model is built; and (ii) absence of consideration of water potential in wood formation processes.

Thank you for these comments. We chose to focus on what we believe to be a fundamental biological question, namely the causes of, and controls on, typical conifer tree-ring anatomies in temperate and boreal climates. We tested the potential for seasonality in developmental zone widths and temperature, along with the spatial gradient in the supply of carbon, to explain observed wood anatomy and the mechanism(s) behind the temperature signal in maximum latewood density. We were also looking to develop a mechanistic sub-model of growth for inclusion in whole-tree and global vegetation models. Carbon supply and temperature are two of the most important components of environmental change, and so we expect our model will make a significant contribution to our understanding of how tree growth and carbon sequestration will evolve over the coming decades. We were able to manipulate our model to gain important insights into the potential mechanisms explaining key phenomena.

We very much agree that water potential can play an important role in wood development, and indeed have recently published a paper on this (Eckes-Shephard, A.H., Tiavlovsky, E., Chen, Y., Fonti, P. and Friend, A.D., 2020. Direct response of tree growth to soil water and its implications for terrestrial carbon cycle modelling. *Global Change Biology*, 27, 121-135. doi:10.1111/gcb.15397). However, adding variability in hydrological effects to our study would have complicated an already complex analysis of mechanisms. A key feature of our model is that the enlargement rates of individual cells drive cell proliferation and interact with development zone widths and carbohydrate supply to produce final cell sizes and density distributions. While we have only considered the influence of temperature on enlargement rate so far, we hypothesise that other factors, such as water potential and nutrients, influence wood formation through this same route (we mention this on lines 288–289 and have also added more on this in lines 83–86). Subsequent work will examine the ability of this assumption to explain responses to water potential, nutrients, photosynthesis, etc., but in each case keeping the fundamental structure the same as presented in this manuscript.

The reviewer questions the generality of our approach. We chose *Pinus sylvestris* to inform our model development due it being well-researched and we therefore had access to numerous and diverse observations, used for direct parameterisation, calibration, and to test the validity of the fundamental model structure. To our knowledge a comprehensive set of all such data types for other species is lacking. We used observations from two sites to determine model parameters using calibration (Table 1) and direct parameterisation (Tables 2 and 3), and found that the model then reproduces observations at more southerly sites (Figure 2a). Now that the fundamental structure has been validated for this species, the model can be adapted to other species using minimal additional observations. The parameters in Table 1 could be re-calibrated for any site, and so the model is completely general (at least for conifers). For example, while the underlying assumption that daylength controls the seasonal progression of zone widths is general, it is likely that the response of zone widths to daylength varies with latitude as a result of local adaptation.

Regarding transferability of the model to other sites and consideration of water potential, we agree completely that consideration of water supply effects will be necessary to accurately simulate wood formation at water-limited sites, and nutrient constraints will also need to be included. We will be incorporating these dependencies in subsequent versions of our model through effects on the rate of the rate of enlargement. We have added text to better explain why our model is able to provide major new general insights into wood formation despite its current lack of a water potential response: lines 80–86.

The author claims (L203-206) that his model could serve as a basis for representing wood formation in TEMs. This is desirable indeed, but considering that TEMs do not include explicit representations of phloem fueling, I would also expect the author to give leads about how to adapt and implement his model in TEMs.

We discussed how our approach could be used in global vegetation models in our paper: Friend, A.D., Eckes-Shephard, A., Fonti, P., Rademacher, T.T., Rathgeber, C., Richardson, A.D. and Turton, R.H., 2019. On the need to consider wood formation processes in global vegetation models and a suggested approach. *Annals of Forest Science*, [doi:10.1007/s13595-019-0819-x](https://doi.org/10.1007/s13595-019-0819-x). We include a reference to this work in the Conclusions (line 291).

Specific comments:

L77-78: recently Delpierre et al. (2019) published parameterizations for such models on various conifers, among which Pinus sylvestris.

Thank you for pointing this out. In fact we tested the CiHS model of Delpierre *et al.* (2019) at the two sites used in our study, with parameter values for *P. sylvestris* taken from their Table S2. Unfortunately their model with these parameter values failed to predict the initiation of cambial activity sufficiently early. We therefore decided to use the original Cannell and Smith (1983) phenology model, setting the threshold temperature to 0°C, and then adjusting the asymptote to produce cambial initiation on DOY 112 (22nd April, 1995) at the observational site used in Uggle *et al.* (2001), as described in the manuscript. Breaking of dormancy on this day is necessary for wood formation to have proceeded to the state observed on DOY 185 in Uggle *et al.* (2001). Confirmation of this parameterisation occurred when the model was then transferred to the Torneträsk site, and predicted the observed absolute diameter growth and its inter-annual variability well (the latter following calibration of E_a), as shown in Figure S3.

L81: remove “highly”. “Are plausible” is enough.

Done.

Figure 1 caption (and other figures using running means): I’m unsure what “0.06-fraction running mean over 0.02-fraction bins.” Mean. Please rephrase.

We have rephrased this caption and similarly changed all other relevant captions to make them clearer.

Figure 2 inset: Unsure i understand the y-scale unit. It does not correspond with the magnitude of change observed on the main graph of Fig. 2 (e.g. on the inset, the sensitivity under the condition “saturating carbohydrates” is about 0.10 g/cm³/K, when on the main figure we switch, e.g. at 750 μm from the cambium from 0.5 g/cm³ (default condition) to 1.1 g/cm³ (saturating CH condition) for a +2K increase: I expect a sensitivity of (1.1-0.5)/2 = 0.30 g/cm³/K).

The “saturating carbohydrates” response in the inset was calculated as the difference between two simulations at different temperatures, but both with saturating carbohydrates, rather than a difference with and without saturating carbohydrates. We have re-drawn this figure and re-written its caption to make it clearer.

L157: “shown”

Thank you for spotting this - corrected.

L206: another paper of interest is Guillemot et al. (2017)

Thank you for this suggestion. We have added this reference.

Table 3: How was the DoY 185 limit set? Arbitrarily? Summer solstice is on DoY 172, so one cannot argue that this is a “natural” limit.

This is the first DOY (i.e. 4th July, 1995) in the measurements of Ugglå et al. (2001), and as such is the first of the three dates through which the widths of the developmental zones versus daylength are fitted (see section on “Zone widths”; Figure S2 shows the assumed changes in zone widths prior to the first observational date).

Equations 11 and 14: you have to multiply (not divide) by 10¹² to convert μm³ (= (10⁻⁶)³ = 10⁻¹⁸ m³) to mL (= 10⁻⁶ m³)

This factor converts from length³ in μm to volume in ml and is correct:

$$\text{ml} = 10^{-6} \text{ m}^3$$

$$\mu\text{m}^3 = 10^{-18} \text{ m}^3$$

$$10^{-18} = 10^{-6} / 10^{12}$$

L272: “the phloem carbohydrate is prescribed”: how is it prescribed? Constant along the season? Varying? Based on what data?

The phloem carbohydrate concentration is fixed at the value given in Table 3 (i.e. 95 mg ml⁻¹). We have added this information to the text referred to make this clearer: ‘The carbohydrate concentration in the phloem is prescribed at the mean value observed across the three observational dates in \cite{uggla_function_2001}, as described below in “Observations” (lines 362–364).

Equation 17: what is the applicability range of this parameterization? whole distribution area of *Pinus sylvestris*? Site-specific?

This cannot be inferred with certainty from our study as it is based on a calibration to one site. This parameterisation appears to be valid at the second site, Torneträsk, which has significantly different daylength seasonality. However, as alluded to above, it is very likely that different provenances/genotypes will have evolved to their local conditions, and so the effect of daylength on developmental zone widths will be something that varies throughout the geographical range. This will be incorporated into applications of our approach when scaled up (and indeed such a local adaptation approach was already used in earlier work with regard to the synchronising of foliar phenology with local climatic conditions through a teleological approach: Friend and White, 2000. Evaluation and analysis of a dynamic terrestrial ecosystem model under preindustrial conditions at the global scale. *Global Biogeochemical Cycles* 14, 1173-1190).

L296: when you refer to dormancy here, i assume this is dormancy of the cambium (i.e. no more cell division in the proliferation zone). This is unclear.

We have clarified that this refers to “cambial”, line 391.

L351: “Wood formation in trees” (not “gormation”)

Thank you for spotting this - corrected.

References cited:

Cabon, A., Peters, R. L., Fonti, P., Martínez-Vilalta, J., & De Cáceres, M. (2020). Temperature and water potential co-limit stem cambial activity along a steep elevational gradient. *New Phytologist*, 226(5), 1325-1340.

Delpierre, N., Lireux, S., Hartig, F., Camarero, J. J., Cheaib, A., Čufar, K., ... & Huang, J. G. (2019). Chilling and forcing temperatures interact to predict the onset of wood formation in Northern Hemisphere conifers. *Global change biology*, 25(3), 1089-1105.

Guillemot, J., Francois, C., Hmimina, G., Dufrêne, E., Martin-StPaul, N. K., Soudani, K., ... & Delpierre, N. (2017). Environmental control of carbon allocation matters for modelling forest growth. *New Phytologist*, 214(1), 180-193.

Lockhart, J. A. (1965). An analysis of irreversible plant cell elongation. *Journal of theoretical biology*, 8(2), 264-275.

Vaganov, E. A., Hughes, M. K., & Shashkin, A. V. (2006). Growth dynamics of conifer tree rings: images of past and future environments (Vol. 183). Springer Science & Business Media.

Reviewer #2 (Remarks to the Author):

Revision and comments on the manuscript “Wood structure explained by complex spatial source-sink interactions” by Andrew D. Friend

The main problem this manuscript address is the lack of understanding of how wood structure in Scots pine is formed. The research uses a sophisticated model to estimate wood structure characteristics and how and why it predicts the formation and transition from Earlywood to latewood. One of the main conclusions highlighted is that the distribution of carbohydrates available influences these wood density variations during xylogenesis, which is very interesting. However, I could not understand from the manuscript how the model operates, besides reading through the 12 assumptions listed in the text. I think this paper is not acceptable in its current stage because this kind of sophisticated modeling should be much better explained than in this type of short format.

We apologise that the reviewer could not understand how our model operates from our manuscript. In order to further aid comprehension we have added a new conceptual figure (Figure 1) and expanded a paragraph in the main text (lines 102–123). We have also made adjustments throughout the text to aid clarity (these can be seen as edits in bold in the manuscript text (Word) file).

From what I understand from the readings, this manuscript concludes that the wood formation of conifer spp, “now generalized,” depends on the interaction of many factors, which in my opinion, is not scientifically satisfying, at least how is presented now. This statement might sound subjective, but I try to explain in the following paragraphs with the best of my intentions. I think it all comes down to the focus and structure of the manuscript that leaves me with this idea.

The focus of the manuscript and claims are based on the results are depending heavily on the model output. The model has a high level of parametrization and an extensive list of assumptions. I am not saying this is inherently bad, but this should be much better explained, in detail, so there are no loose ends, or at least those loose ends are well addressed.

Indeed, our results are model-output dependent. This is a theoretical study, examining potential mechanisms behind key observed patterns. The model outputs are primarily the result of the assumptions which, while numerous, are based on observational/experimental studies and biological theory. Our study tests the plausibility of the collection of these individual assumptions acting together in one framework. The different model simulations are designed to explore the mechanisms responsible for observed patterns. These assumptions interact within our model to produce realistic behaviour and provide major new insights. The success of our model in reproducing and explaining a range of key features in real tree rings is very exciting, and we believe indicates real insight into how xylogenesis operates (potentially in all trees).

We have kept the number of parameters to the absolute minimum and the majority are derived directly from observations independently of our model. Six are calibrated (i.e. determined so that the model behaves as observed: Table 1). Our model and its

number of free parameters are typical of wood formation modelling frameworks (see Schliestl-Aalto et al, 2015 and Drew et al 2010). The additional text noted above and the various edits we have provided throughout provide a fuller and clearer explanation of how we have developed, parameterised, and tested our model.

One of the claims in the article, states that this model can help improve the carbon cycle models is a bit of a long shot because it would be impossible or at least not reasonable to extrapolate this model to all tree spp. and climates of the world. At least this is how it looks from the narrative. Also, it is not explicit how this approach can help the land surface models that the author refers to? One of the concerns I have, is how the author can justify those claims when knowing that there are considerable variations in xylogenesis and wood structure across the world? In the text of the manuscript is not explicitly stated how this model is extrapolable to other species.

Regarding the comment on land surface models, as mentioned in our response to Reviewer #1, a previous publication (by us) includes a detailed analysis how a model of xylogenesis like ours can help improve carbon cycle models (i.e. Friend *et al.*, 2019. On the need to consider wood formation processes in global vegetation models and a suggested approach. *Annals of Forest* 76, 49), and we refer to this in the text (ref. 5).

We fully acknowledge that xylogenesis and wood structure vary considerably across species, and that our model has so-far only been parameterised and tested for one, Scots pine. However, what we find exciting and show in our article is the model's ability to reproduce the key features of wood formation and explain the resulting distribution of density through the ring in this species, as well as its sensitivity to temperature. These findings give confidence that our model captures fundamental processes that potentially operate in many (all?) tree species (e.g. interactions between the widths of proliferation, enlarging, and wall thickening developmental zones and the supply of carbohydrates. We acknowledge that this needs to be tested, and is the focus of other work we are involved with, including experiments at Harvard Forest on red oak, red maple, and white pine (e.g. Chen *et al.*, 2022. Inter-annual and inter-species tree growth explained by phenology of xylogenesis. *New Phytologist* 235, 939-952.), and a recent collaboration with the Sainsbury Laboratory University of Cambridge and University of Helsinki to directly test and extend our model to angiosperms using hybrid poplar as a model species (<https://gtr.ukri.org/projects?ref=NE%2FW000199%2F1#/tabOverview>). We have added text to the conclusions discussing extension of our model to other species and refer to the model of Drew *et al.* (2010) as an example of an approach to simulate cell types specific to angiosperms (lines 291–293).

Another aspect that I find weak is that the methods in this manuscript are set on a lower profile (less critical) when I think this should be a powerful factor on this type of manuscript. I think that the details on the description of the “observations” should play a much bigger role in the paper's main body and explain with much more details and clearer language.

As previously mentioned, we have now added more details on how the model operates and have made edits throughout to improve clarity. The model is already fully described in the Methods, where all equations and parameter values are given.

The section on “Observations” explains how these existing observations were used to derive the model parameters from the paper of Uggla *et al.* (2001) and the source of the calibration targets for the temperature sensitivities (i.e. Grudd *et al.*, 2008). We have added some further details in lines 424–427 and 445–448, but do not believe that our paper would benefit from greater discussion of these sources as they are fully described elsewhere. The focus of our article is the generation of new knowledge through simulations with our model framework under different assumptions, rather than calibration to any particular dataset.

The results and figures are poorly presented, and the figures could be more carefully produced, with higher quality, so it is easy for the reader to follow the main messages.

Thank you for this observation. In response we have re-drawn all of the figures to improve their clarity and refer more explicitly to their content throughout the text.

Additionally, the narrative of the paper is a bit complicated to follow. Many important concepts in the manuscript are not well introduced, for example; 1) The developmental stages,

As mentioned, we have edited the text to improve clarity in many places, including with regard to the developmental stages (e.g. fifth paragraph of Main, lines 102–123).

2) The observations made on *Pinus silvestris* we are left with only the fact the author made “critical observations” that is all until we arrive in the figures with plenty lines showing the “observations”.

We did not create any observations in this study, and therefore this comment is not clear to us.

4) In the last sentence before the conclusions, there is a statement with the “divergence problem” which has not been properly introduced or addressed throughout the manuscript.

We have added text to the Main section on the significance of the divergence problem (lines 38–42).

Reviewer #3 (Remarks to the Author):

The manuscript “Wood structure explained by complex spatial source-sink interactions” presents a model of cell growth in cambium to form a framework to explain possible mechanisms to form wood of a tree so as to attribute tree ring density profile. The well known theory, observation and suitable assumption are employed to make simulation test. And significant fresh and new ideas are addressed: NSC diffusion and its responses to temperature is a dominant reason to determinate the tree ring density (hence wood volume mass content). From this idea, the present time' volum based tree carbon storage computing should be updated by considering temperature effect on tree ring density (via earlywood and latewood). Therefore this paper will be focused on by many researchers even common readers. I think this paper should be accepted to publish.

Thank you very much for your positive assessment of our work - this is greatly appreciated.

suggestions:

The concept and structure of this model could be clearer if there is a figure to present the whole framework.

Thank you for this suggestion. As mentioned in our response to Reviewer #2 above, we have added an overview figure of the model (Figure 1) and refer to this in new text (lines 102–123).

This model does not simulate the NSC dynamics (but uses the observed NSC). Recently, some studies have constructed the physiological models that modeled the NSC dynamics and cell growth, e.g. the CASSIA model (Schiestl-Aalto et al. (2015)) and FORCCHN2 model (Fang et al. (2020)). these references's findings and ideas may strengthen the understanding of this work.

Thank you for these suggestions. However, for this particular study it is not clear to us how to relate CASSIA and FORCCHN2 to our model as they are rather empirical and compute whole-plant NSC, whereas our model assumes a constant phloem concentration of NSC as a source of sugars for the developing radial file. However, these suggestions are valuable for us to keep in mind as we develop our whole-tree model (as described in Friend et al., 2019).

Specific comments:

Fig. 1: related questions.

Sweden site modeled data were comparison to other 4 sites's observation, which may be questioned. Because this paper regarded climate's effects (temperature) as important factors, those other 4 sites climate condition need to be compared to Sweden site. My understanding is that except for the saturating experiment run, the carbohydrates always keep as observation. If so, it seems the single ring mass integration should be almost equal in different experiment runs. However from figure, it seems that fixed zone, fixed temperature, no dormancy produce every different mass amount in a same tree ring.

While the carbohydrate concentration in the phloem is fixed at its observed value (Uggla et al., 2001), the carbohydrate concentration profile across each radial file varies as a function of the demand for wall growth by each cell, with an equilibrium gradient calculated for the conditions on each day. The supply of carbohydrates to each cell therefore varies according to this gradient. This creates a complex spatial interaction between demand and supply of carbohydrates, which varies over the year as temperature and zone widths change. Changes to the assumptions of the model have profound effects on these interactions and, as a consequence, on the resulting annual radial profiles of wood anatomy.

The example referred to by the reviewer can be explained by the fact that under fixed zone widths, cell production and enlargement-only zones are proportionally larger

than under declining zone widths. Similarly, cell production and enlargement are prolonged under runs without dormancy. One might first expect that in these cases more cells are produced, and more carbon would get deposited. However, these increased number of cells simultaneously do not get the chance to incorporate large amounts of carbon in their secondary cell walls, as their duration in the thickening zone is reduced (no dormancy), or the last cells are never able to enter the thickening zone (constant zone width, no dormancy) and therefore a different total mass is deposited. We have added text to make this aspect of the model clearer (lines 173–177).

Fig. 2 related questions.

+2K experiment runs may increase ring width(from fig) ?
only wall thickening experiment run seem to allocate more mass in tree ring, however carbohydrate amount should be fixed to same value if my understanding is right.

As stated in our previous response, while the phloem concentration is fixed, the amount of supply will vary as a function of demand. Hence higher temperatures can draw more carbohydrate into the developing radial file, and so more mass and volume are produced. We have adjusted the wording to make this clearer (lines 236–240).

Table1: μ_0 is the final ring width? In this work, the μ_0 is a relative value, and it is equal to $0.0559 \mu\text{m} \mu\text{m}^{-1}\text{day}^{-1}$. E_a is the ring width response to temperature? The E_a may represent the density response to temperature.

μ_0 is the relative radial growth rate of an individual cell at T_0 . E_a is the activation energy for the rate of enlargement of an individual cell. The density response to temperature is an outcome of the model, not a parameter.

Line 126-127: the density distribution was determined by the activities of cambium via NSC. beside the temperature' effect, hydraulics from xylem also influence the density distribution.

Indeed, water availability would impact wood formation at sites with water limitation. The model, being developed at a temperature-only limited site, does not currently include an effect of hydraulics, but doing so would be compatible with its structure. We have added text explaining this on lines 83–86.

Line 323: The observations from the published paper, and you could add a simple description of the observations.

We have added some details as to how the observations were obtained in the Uggla *et al.* and Grudd *et al.* papers to the section “Observations”, i.e.: “Samples were cut from three 44 yr old Scots pine trees growing in Sweden (64° 21' N; 19° 46' E) at different times during the growing season. 30 μm thick longitudinal tangential sections of the cambial region were made, and the radial distributions of soluble carbohydrates measured using microanalytical techniques (Uggla *et al.* 2001).” (lines 424–427) and “These observations were made on living and subfossil Scots pine sample material from the Lake Tornestrask

area (68.21–68.31 \degree N; 19.45–19.80 \degree E; 350–450 m a.s.l.) using X-ray densitometry for maximum density, and standardised to remove non-climatic information\cite{grudd_tornetrask_2008}.” (lines 445–448).

Reviewer #2 (Remarks to the Author):

Review of the manuscript titled Wood structure explained by complex spatial source-sink interactions.

I am happy to read this new version of the manuscript. Furthermore, I believe this version is worth publishing in the current version because the authors have addressed the most relevant and essential aspects that the first reviewers have addressed. Those changes and other additional details added to this new and much better version have made this article much better, and I think it is well structured and relevant to the readership in this journal.

I think it is an excellent contribution to our understanding of wood formation. I think it will resonate with many authors thinking and making many of the assumptions of this model, and the work done here, providing this type of framework, will be very useful for future efforts in the field.

Reviewer #3 (Remarks to the Author):

ok